# Chemokine Receptors in Peripheral Blood Mononuclear Cells as Predictive Biomarkers for Immunotherapy Efficacy in Non-Small Cell Lung Cancer

**DOI:** 10.3390/curroncol32100583

**Published:** 2025-10-20

**Authors:** Paloma Galera, Antía Iglesias-Beiroa, Berta Hernández-Marín, Dulce Bañón, Teresa Arangoa, Lucía Castillo, María Álvarez-Maldonado, Cristina Gil-Olarte, Rafael Borregón, María Iribarren, Ramon Colomer, Jacobo Rogado

**Affiliations:** 1Medical Oncology Department, Hospital Universitario de La Princesa, Diego de León 62, 28006 Madrid, Spain; antiaiglesiasbeiroa@gmail.com (A.I.-B.); bertahernandezmarin@hotmail.com (B.H.-M.); dulce.banontorres@gmail.com (D.B.); m3arang@yahoo.es (T.A.); l.castillofdez@hotmail.es (L.C.); mamaldonadotp@gmail.com (M.Á.-M.); crigilmon@gmail.com (C.G.-O.); rafael.borregon@gmail.com (R.B.); miribarrenvaler@gmail.com (M.I.); rcolomer@seom.org (R.C.); 2Instituto de Investigación Sanitaria La Princesa, Diego de León 62, 28006 Madrid, Spain; 3Department of Medicine and Chair of Personalized Precision Medicine, Universidad Autónoma de Madrid, 28006 Madrid, Spain

**Keywords:** immunotherapy, non-small cell lung cancer, chemokine receptors, CXCR4, CXCR3, CCR5, CXCR6, predictive biomarkers, immune checkpoint inhibitors

## Abstract

**Simple Summary:**

Immune checkpoint inhibitors (ICIs) have improved survival in selected patients in Non-small cell lung cancer (NSCLC), yet reliable predictive biomarkers are still lacking. Peripheral blood mononuclear cells (PBMCs) provide a minimally invasive tool to evaluate systemic immunity and may reveal such biomarkers. Chemokines and their receptors, particularly CXCR4, play crucial roles in immune regulation and antitumor responses. Their expression on PBMCs has been linked to better immune cell trafficking (better migration of T cells, i.e.,) and ICI efficacy. This review summarizes current evidence on PBMC chemokine receptor expression as a potential predictive biomarker in NSCLC immunotherapy.

**Abstract:**

Non-small cell lung cancer (NSCLC) remains a leading cause of cancer-related mortality globally. The advent of immune checkpoint inhibitors (ICIs) has significantly improved outcomes for a subset of patients; however, predictive biomarkers to identify responders are still lacking. Peripheral blood mononuclear cells (PBMCs) offer a minimally invasive means to assess systemic immune status and have emerged as a potential source of predictive biomarkers. Recent studies have highlighted the role of chemokines and their receptors in modulating immune responses against tumors. In particular, the expression levels of chemokine receptors such as CXCR4 on PBMCs have been associated with patient responses to ICIs. The differences in expression of these receptors correlates with enhanced T cell trafficking and infiltration into the tumor microenvironment, leading to improved antitumor activity. This review consolidates current evidence on the prognostic and predictive value of chemokine receptor expression in PBMCs, like T cells, for NSCLC patients treated with ICIs. Understanding these associations can aid in the development of non-invasive biomarkers to guide treatment decisions and improve patient stratification in immunotherapy.

## 1. Introduction

Lung cancer is the second most common malignancy and the leading cause of cancer-related death among both men and women [1]. Non-small cell lung cancer (NSCLC) constitutes approximately 80% of all lung cancer cases [2]. Despite advancements in multiple approaches and therapies such as chemotherapy, targeted therapies, immunotherapies and radiotherapy, the five-year survival rate for NSCLC patients remains poor, particularly in advanced stages [3]. Most patients are diagnosed with advanced or metastatic disease, limiting the effectiveness of traditional treatments [4]. Therefore, there is an urgent need for novel therapeutics and improved strategies to enhance existing treatments [5].

In recent years, Immune Checkpoint Inhibitors (ICIs) anti-PD-1 (programmed death 1) or anti-PD-L1 (programmed death-ligand 1) antibodies, have shown promising results in clinical trials and are approved for the treatment of advanced NSCLC. These therapies aim to activate the patient’s immune system to identify and destroy cancer cells, by the inhibition of the immune self-tolerance [6,7,8,9].

However, the efficacy of ICIs is variable, with overall response rates remaining relatively low (15–50%) and the duration of response often limited to about 12 months. Most patients either show primary or acquired resistance to ICI treatment, and non-responders can suffer from severe immune-related adverse events [5,10]. Currently, PD-L1 expression on tumor cells, assessed by immunohistochemistry, is the most extensively used predictive biomarker for ICI administration in NSCLC patients. While high pretreatment PD-L1 expression often correlates with superior clinical outcomes, its application as a predictive biomarker has deficiencies. Many patients with PD-L1-negative tumors respond to therapy, and there are challenges related to intratumoral heterogeneity, varying expression levels between primary and metastatic lesions, lack of standardization in detection methods, and the dynamic nature of PD-L1 expression. Tumor Mutational Burden (TMB) has also been explored, but its predictive value remains controversial, and it faces limitations such as high cost, long turnaround times, and lack of standardization [2,6,11].

Given these limitations, there is a crucial need for additional predictive biomarkers to accurately stratify patients, select appropriate treatments, and monitor responses in real-time. Peripheral blood-based biomarkers are gaining increasing interest due to their non-invasive nature, ease of access, and potential for repetitive sampling, offering a holistic insight into the host immune status. Among various blood-based markers, chemokines and their receptors play a pivotal role in regulating immune cell trafficking and modulating the tumor microenvironment (TME) [10,11,12]. This review will focus on the emerging role of chemokine receptors expressed on peripheral blood mononuclear cells (PBMCs) as predictive biomarkers for ICI efficacy in NSCLC, consolidating current evidence and discussing their implications for personalized cancer therapy.

## 2. Non-Small Cell Lung Cancer: Epidemiology, Classification, and Current Treatment Approaches

Lung cancer remains a formidable global health challenge, with significant morbidity and mortality [13,14]. Environmental factors, primarily cigarette smoke and radon exposure, are the major risk factors, alongside genetic alterations. In 2018, it was estimated that over 230,000 new cases of lung cancer would be diagnosed in the United States, leading to over 150,000 deaths. While incidence rates have declined in men due to changes in smoking behavior, the incidence is increasing in women. A notable trend is the increased frequency of lung cancer among non-smoking young women, though the causes are not fully understood [3,5].

### 2.1. Classification of NSCLC

The World Health Organization (WHO) revised its classification of lung tumors in 2015, defining adenocarcinomas, squamous cell carcinomas, and neuroendocrine tumors as the three major types of malignant tumors of the lung [15]. Lung adenocarcinoma is the most common histological subtype globally, accounting for almost half of all lung cancers. [5].

### 2.2. Standard of Care for Advanced NSCLC

Historically, effective treatment options were limited for NSCLC patients without actionable driver mutations who experienced disease progression with a very limited survival [16]. Even with targeted therapies for patients with oncogenic alterations (such as activating mutations in *EGFR*, *cMET*, *ALK*, *BRAF*, *NTRK*, *KRAS*, and *ROS-1*), metastatic NSCLC remains incurable [17].

Current first-line treatment for patients with metastatic non-small cell lung cancer who have not previously received treatment and without any molecular alterations includes: platinum-doublet chemotherapy with or without bevacizumab for those with nonsquamous cancer, anti-programmed death 1 (*PD-1*) monotherapy for those with programmed death ligand 1 (*PD-L1*) expression on at least 50% of tumor cells and anti-*PD-1* (+/− *anti-CTLA4* combinations) plus platinum-doublet chemotherapy, treatment that is the current standard [18,19,20,21].

### 2.3. Emergence of Immunotherapy and Its Limitations

Immune Checkpoint Inhibitors (ICIs) have significantly improved cancer treatment efficacy in NSCLC. Consequently, ICIs are being adopted as first-line therapeutic options [22,23]. Recently, these therapies have also been indicated in the neo-adjuvant, adjuvant o peri-adjuvant setting. However, despite their promise, ICIs have significant limitations: overall response rates are relatively low (15–50%), duration of response is limited (around 12 months) and there are not efficacy biomarkers beyond *PD-L1* tumor expression [10].

### 2.4. Challenges with Current Biomarkers

The efficacy of *anti-PD-1* immunotherapy in NSCLC patients is variable, and reliable biomarkers beyond tumor *PD-L1* expression in biopsy are lacking. *PD-L1* expression itself is considered a heterogeneous and dynamic biomarker. Furthermore, many patients with *PD-L1-negative* tumors may still respond to *anti-PD-1/PD-L1* therapy. Biological limitations, such as intratumoral heterogeneity, dinamic expression levels between primary and metastatic lesions, and lack of standardization in detection antibodies and scoring systems, complicate its utility [24,25].

Tumor Mutational Burden (TMB) is another emerging predictor but has yielded controversial results [10].

These challenges highlight the need for new, more reliable, and less invasive biomarkers. Peripheral blood is a highly attractive source for biomarkers due to its easy accessibility, minimal invasiveness, and suitability for repeated sampling, which can provide a comprehensive insight into the host immune status [26].

## 3. Chemokines and Their Receptors in Cancer Progression and the Tumor Microenvironment

Chemokines are a superfamily of small molecule chemoattractive cytokines that mediate various cellular functions. They are crucial for guiding immune cell homing signals in the body under different conditions [3,27,28]. However, in cancer, tumor cells could utilize it to promote local tumor growth and distant spread [29]. Chemokine abnormalities can disrupt homeostasis and immune function, contributing to cancer development stages such as invasion or promotion of metastasis [30].

There are approximately 50 types of chemokines, and their receptors can interact in an autocrine or paracrine manner, aiding tumor proliferation, survival, and neo-angiogenesis. Distant metastasis is sometimes guided by organs expressing specific chemokines, which attract tumor cells expressing the particular chemokine receptors towards these specific organs [3].

The tumor microenvironment (TME) is a complex ecosystem comprising cancer cells, diverse immune cells, fibroblasts, endothelial cells, and extracellular matrix components [31,32,33]. Chemokines play a multifaceted role within the TME, influencing tumor cell proliferation and survival, angiogenesis (formation of new blood vessels), metastasis and invasion, immune cell infiltration and modulation of immune responses, cancer stem cell self-renewal and chemo-resistance and inflammation [4].

Several atypical chemokine receptors or decoy receptors (ACKRs) also exist, which internalize and degrade ligands, thereby “cleaning them up” from the microenvironment. Examples include *ACKR1 (DARC)*, *ACKR2 (D6-Duffy antigen)*, *ACKR3 (CXCR7)*, and *ACKR4 (CCLR1)*. *ACKR3 (CXCR7)* is overexpressed in several tumors and drives tumor cell adhesion, invasion, survival, and growth, as well as angiogenesis [3,34,35]. In NSCLC, *CXCR7* overexpression in tumor cells promoted primary tumor growth and metastasis in A549 cells, but its silencing inhibited *TGFβ1-induced* migration, invasion, EMT, and reduced stem-like properties and chemoresistance [30].

This intricate network of chemokines and their receptors significantly impacts cancer progression and patient outcomes, making them potential targets for diagnostic and therapeutic interventions [3,5].

## 4. Chemokine Receptors as Prognostic and Predictive Biomarkers in NSCLC

### 4.1. CXCR4 (C-X-C Chemokine Receptor Type 4) Is a Candidate Oncogene in Several Human Tumors, Including NSCLC

It is expressed in over 30 types of malignant tumors [36,37]. Its ligand is *CXCL12* (also known as stromal cell-derived factor-1 or *SDF-1*). The interaction of *CXCL12* and *CXCR4* has a broad impact on tumor cell proliferation, survival, angiogenesis, metastasis, and the tumor microenvironment [30,38,39].

#### 4.1.1. Pathological Roles of *CXCR4* in NSCLC

Extensive research and multiple meta-analyses have explored the multifaceted roles of the chemokine receptor *CXCR4* in non-small cell lung cancer (NSCLC), identifying it as a key player in several tumor-promoting processes [40,41]. *CXCR4* exerts pleiotropic pro-tumorigenic effects, particularly in supporting tumor growth and proliferation. It enhances anti-apoptotic activity primarily through activation of the *JAK2/STAT3* signaling pathway, which in turn reduces the efficacy of chemotherapeutic agents such as cisplatin, thereby contributing to treatment resistance [30,38,42].

In addition to its role in tumor growth, *CXCR4* is critically involved in tumor metastasis and invasion. High levels of *CXCR4* expression in the cytomembranous compartment of lung cancer cells have been correlated with increased local invasion and distant metastasis, particularly to the brain and pleural space. Mechanistically, *CXCR4* facilitates actin polymerization and pseudopodia formation, which mediate chemotactic and invasive behavior. Furthermore, environmental factors such as hypoxia and the presence of angiogenic signals can upregulate *CXCR4* expression, further promoting metastatic potential [43,44,45].

*CXCR4* expression also shows a strong association with lymph node metastasis. Elevated levels of *CXCR4* in tumor cells are frequently accompanied by high expression of its ligand *CXCL12* within tumor-harboring lymph nodes. This suggests a directional migration mechanism in which high *CXCL12* concentrations in lymph nodes attract CXCR4-expressing cancer cells, thereby facilitating lymphatic spread [5,46].

Another key function of *CXCR4* in NSCLC is its role in maintaining cancer stem cell (CSC) populations. In A549 NSCLC cell lines, *CD133^+^CXCR4*^+^ cells exhibit enhanced migratory capacity and features consistent with epithelial–mesenchymal transition (EMT). These findings indicate that *CXCR4* contributes to the self-renewal and aggressive behavior of CSCs and mediates *CD133-induced* migratory and EMT-associated phenotypes [44,47].

Moreover, *CXCR4* plays a role in modulating the tumor immune microenvironment, contributing to immune suppression. It facilitates the recruitment of proinflammatory cells that may help establish an immune-permissive tumor microenvironment (TME), which could impair antitumor immune responses and affect the efficacy of immunotherapies [2,5].

Finally, *CXCR4* expression has been correlated with both disease stage and tumor histology. Higher expression levels of CXCR4 are more frequently observed in patients with advanced-stage tumors and are particularly associated with adenocarcinoma histology, underscoring its relevance as a potential diagnostic and prognostic marker in NSCLC [5,48].

#### 4.1.2. *CXCR4* as a Prognostic Marker

Multiple studies, including comprehensive meta-analyses, have established *CXCR4* as a clinically relevant prognostic marker in NSCLC [37,40,49]. High *CXCR4* expression in resected NSCLC tumors has been consistently associated with poor overall survival [17,48,50]. In patients with stage IV NSCLC, those exhibiting high CXCR4 expression showed a significantly reduced median overall survival of 2.7 months, compared to 5.6 months in patients with low expression levels [17].

Interestingly, the negative prognostic impact of *CXCR4* overexpression appears to be influenced by gender. Specifically, female patients with stage IV NSCLC and high *CXCR4* expression experienced notably poorer clinical outcomes, with a median overall survival of only 1.6 months, in contrast to 6.4 months for low expressors. This observation suggests a potential gender-dependent difference in the prognostic implications of *CXCR4* [17]. In support of this, further studies have quantified *CXCR4* expression in early-stage disease and investigated its association with gender-specific recurrence-free survival and overall survival outcomes [51]

However, the prognostic relevance of *CXCR4* localization within the cell remains a subject of debate. While strong cytomembranous expression of *CXCR4* correlates with worse prognosis, other studies have reported that nuclear staining of *CXCR4* in tumor cells may be linked to improved clinical outcomes. Despite these findings, the association between nuclear localization and favorable prognosis has not been consistently replicated [5,52,53]. Discrepancies in antibody specificity may underlie these conflicting results [17]. For instance, the *UMB-2* rabbit monoclonal antibody, known for its high specificity in detecting membrane-bound *CXCR4* with minimal nuclear staining, aligns more closely with the established function of *CXCR4* as a membrane receptor, casting doubt on earlier nuclear staining interpretations [54].

#### 4.1.3. *CXCR4* as a Therapeutic Target

Given its central role in cancer progression and metastasis, *CXCR4* has emerged as a highly promising therapeutic target. Currently, more than 15 agents designed to disrupt the *CXCR4–CXCL12* interaction are under various stages of development, reflecting substantial interest in this axis as a focus for drug Discovery [3,55].

Among the *CXCR4* antagonists, AMD3100 (also known as plerixafor or Mozobil) is the most extensively studied. Originally developed as an anti-HIV agent, AMD3100 was later approved by the FDA for the mobilization of hematopoietic stem cells in preparation for bone marrow transplantation. It functions by binding to the ligand-binding pocket of *CXCR4*, thereby inhibiting the interaction with *CXCL12* and effectively blocking downstream signaling pathways. In preclinical cancer models, AMD3100 has demonstrated multimodal antitumor effects across a variety of tumor types [3,53,56]. Another notable antagonist, BKT140 (BL-8040), has shown therapeutic efficacy specifically in human NSCLC models, where it inhibited tumor cell proliferation and enhanced the cytotoxic effects of cisplatin, paclitaxel, and radiation therapy [3,57]. Additional antagonists under investigation include LY2510924, LY2624587, Ulocuplumab, and NOX-A12, all of which are being evaluated for their potential to disrupt the *CXCR4–CXCL12* axis in cancer [58].

A key area of research is the use of *CXCR4* inhibitors to enhance the efficacy of immune checkpoint inhibitors (ICIs). The *CXCR4* blockade has been shown to modulate the tumor microenvironment by reducing stromal desmoplasia, facilitating T-lymphocyte infiltration, and alleviating immune suppression—mechanisms that can help overcome resistance to ICIs and improve therapeutic outcomes in various malignancies [5,59]

In addition to its therapeutic applications, *CXCR4* is being explored as a target for diagnostic imaging and endoradiotherapy. Small-molecule *CXCR4* antagonists labeled with radioactive isotopes are under development for the specific in vivo detection of tumors with high *CXCR4* expression. For example, pentixafor, a *CXCR4* antagonist, has been successfully labeled with *68Ga* and *64Cu* isotopes, enabling precise imaging of *CXCR4*-expressing lung tumors. This approach also sets the foundation for targeted radionuclide therapy: when labeled with therapeutic isotopes such as *177Lu* or *90Y*, *CXCR4* antagonists may serve in endoradiotherapy regimens. However, the potential for bone marrow toxicity remains a critical consideration in the clinical application of this strategy [5,60].

#### 4.1.4. *CXCR4* Expression in PBMCs and Immunotherapy Response

Recent studies have specifically investigated the correlation between *CXCR4* expression on circulating immune cells and the efficacy of *anti-PD-1* immunotherapy in NSCLC patients [2,11]. High Circulating *CXCR4*^+^ T cells and Poorer Outcome: A prospective cohort study found that high pretreatment levels of circulating *CD8^+^ CXCR4^+^* T cells correlated with poorer overall survival (22.0 vs. NR months, HR 0.29, *p* = 0.02) in NSCLC patients treated with *anti-PD-1* ICIs. These differences were specific to immunotherapy-treated patients and were not observed in control groups receiving non-immunotherapy treatments [2,11]. Similarly, high levels of *CD4^+^ CXCR4highCD69^+^* T cells accumulate in lung adenocarcinoma [17,61].

Low Circulating *CXCR4*^+^ T cells and Better Outcome: Conversely, another study indicated that low expression of *CXCR4*-expressing CD8^+^ T lymphocytes was correlated with a greater benefit from immunotherapy. This suggests that low circulating *CXCR4^+^ CD8^+^* T cells might lead to increased tumor infiltration by lymphocytes in response to *CXCL12* signaling in tumor cells, creating a proinflammatory environment. This mechanism could contribute to the enhanced efficacy of *anti-PD-1/PD-L1* immunotherapy in NSCLC [2,11,62,63].

This apparent contradiction highlights the complexity of *CXCR4*’s role and its expression patterns in different cellular compartments (circulating vs. tumor-infiltrating) and states (naïve vs. effector T cells). While *CXCR4* on tumor cells promotes progression and metastases, its role on different immune cell subsets in the peripheral blood may have distinct implications for immunotherapy response.

### 4.2. CXCR6 (C-X-C Chemokine Receptor Type 6)

*CXCR6* is another chemokine receptor that plays a crucial role in the tumor microenvironment, particularly in the context of T-cell migration and immunotherapy efficacy. Its ligand is *CXCL16* [64,65].

#### 4.2.1. Role in T-Cell Migration and Function

Resident memory T cells (TRM) represent a specialized subset of *CD8*^+^ T cells that reside permanently within tissues and play a pivotal role in local immunosurveillance, particularly in the context of tumor immunity. *CXCR6* has been identified as a preferentially expressed chemokine receptor on *CD8*^+^ TRM cells following vaccination in murine models, as well as on intratumoral CD8^+^ TRM cells isolated from human lung cancer specimens. The presence or induction of TRM cells has been positively correlated with favorable responses to *anti-PD-1/PD-L1* immune checkpoint blockade, as well as enhanced efficacy of cancer vaccines, highlighting their functional relevance in both natural and therapy-induced antitumor immunity [66,67].

The *CXCR6–CXCL16 axis* is particularly critical for the effective migration and localization of *CD8*^+^ TRM cells within lung mucosa following vaccination. Experimental studies using *CXCR6-deficient* mice demonstrated impaired recruitment of antigen-specific *CD8*^+^ T cells to the lungs, especially affecting TRM subsets. Notably, this migratory defect was linked to the route of vaccine administration: intranasal vaccination, but not intramuscular, induced significantly higher and more sustained levels of CXCL16 in bronchoalveolar lavage fluid and lung parenchyma, thereby promoting optimal TRM positioning and tumor control in pulmonary tissues [66].

Furthermore, *CXCR6^+^ CD8^+^* T cells have been characterized as more immunocompetent compared to their CXCR6^−^ counterparts, further supporting the potential of targeting this axis to enhance immune responses in cancer immunotherapy [64].

#### 4.2.2. *CXCR6* and Immunotherapy Efficacy

*CXCR6* has emerged as a critical regulator of antitumor immunity, with its deficiency significantly impairing the efficacy of immunotherapeutic strategies. In preclinical models, intranasal administration of a cancer vaccine demonstrated reduced antitumor efficacy in CXCR6-deficient mice compared to wild-type counterparts. This reduced response was closely associated with diminished recruitment of local antitumor CD8^+^ tissue-resident memory T cells (TRM). Moreover, *CXCR6*-deficient mice exhibited a markedly diminished response to anti-PD-1 therapy, further underscoring the essential role of *CXCR6* in mediating effective immunotherapeutic outcomes [64,66].

Beyond its necessity for vaccine responsiveness, *CXCR6* also contributes to enhancing the therapeutic impact of immune checkpoint blockade. Induced *CXCR6^+^ CD8^+^* T cells display tumor antigen specificity and are capable of augmenting the efficacy of anti-PD-1 therapy, resulting in slowed tumor progression. Interestingly, high intratumoral expression of *CXCR6* is not primarily driven by chemotactic gradients involving its ligand *CXCL16* but rather is upregulated locally by the tumor microenvironment itself, indicating an intrinsic regulatory mechanism within the tumor milieu [64].

*CXCR6* is also highly relevant in mucosal-associated invariant T (MAIT) cells, a subset of T cells found in the lungs, among other human tissues. These cells express a semi-invariant T cell receptor (TCR) that recognizes microbial riboflavin metabolites presented by the *MR1* molecule, enabling activation and cytotoxic responses [68]. In NSCLC patients who respond to anti-PD-1 therapy, activated and proliferating *CD8*^+^ MAIT cells were significantly enriched and expressed elevated levels of cytotoxicity-related genes. Notably, *CXCR6* expression was markedly upregulated in both circulating and tumor-infiltrating CD8^+^ MAIT cells from responders. These *CXCR6^+^CD8^+^* MAIT cells also showed enhanced expression of genes associated with cytotoxic function, suggesting that *CXCR6* expression may serve as a functional marker of immunotherapy responsiveness. It has been proposed that the immunotherapeutic role of CD8^+^ MAIT cells in lung cancer is mediated, at least in part, by interactions with classical and non-classical monocytes via the *CXCL16–CXCR6* axis [68,69].

Finally, the prognostic relevance of *CXCR6* expression has been increasingly recognized in lung cancer. Loss of *CXCR6* expression impairs the recruitment of *CD8^+^* T cells to pulmonary tissues, contributing to immunological dysfunction. Clinically, low *CXCR6* expression is significantly associated with poor prognosis in lung cancer patients, while higher CXCR6 levels have been linked to improved outcomes, particularly in lung adenocarcinoma. This highlights the potential of CXCR6 as both a prognostic biomarker and a therapeutic target in lung cancer immunotherapy [68,70,71].

### 4.3. Other Chemokine Receptors and Their Significance in NSCLC Immunotherapy

#### 4.3.1. CXCR3

*CXCR3*, along with its ligands *CXCL9*, *CXCL10*, and *CXCL11*, plays a pivotal role in the chemoattraction of effector T cells to the tumor microenvironment. The functional activity of the *CXCR3* chemokine axis within tumors has been shown to be essential for the therapeutic efficacy of *anti-PD-1* immune checkpoint blockade. However, the expression dynamics of its ligands may also carry prognostic significance [72,73]. In a cohort of patients receiving ICI monotherapy, elevated post-treatment levels of *CXCL10* were associated with shorter progression-free survival (PFS) and overall survival (OS), suggesting that sustained high *CXCL10* expression may reflect an ineffective or dysregulated immune response, potentially indicating resistance to therapy. Beyond its role in T-cell trafficking, *CXCL10* mRNA expression has also been proposed as a predictive biomarker of response to neoadjuvant chemoradiotherapy in rectal cancer patients, underscoring its broader relevance across cancer types [72].

#### 4.3.2. CCR5

The chemokine receptor *CCR5* has been implicated in both tumor progression and immune regulation across various cancer types [74]. Its interaction with the ligand *CCL5* promotes the migration and invasiveness of pancreatic cancer cells, and preclinical studies have shown that pharmacological blockade of *CCR5* can significantly reduce tumor growth in murine models of pancreatic cancer [2,75]. Beyond its role in tumor biology, *CCR5* is also essential for the generation of effective T cell–mediated antitumor responses; its expression in both *CD4*^+^ and CD8^+^ T cells is required to achieve maximal immune function within the tumor microenvironment (TME). This dual involvement in tumor progression and immune activity has made *CCR5* an attractive target for therapeutic intervention. In this context, *CCR5* inhibitors are currently being investigated to enhance the efficacy of immune checkpoint inhibitors [51,76].

Notably, the ongoing clinical trial NCT04123379 is evaluating the combination of Nivolumab with BMS-813160—a dual *CCR2/CCR5* antagonist—as a strategy to boost immune responses against non-small cell lung cancer (NSCLC), highlighting the translational relevance of targeting this axis in cancer immunotherapy [3,74].

#### 4.3.3. CXCR2/CXCL5

*CXCR2*, a receptor for chemokines like *CXCL5 (ENA-78)* and *CXCL8* (*IL-8*), is involved in various aspects of cancer biology [3,59].

*CXCL5*: This chemokine contributes to tumor metastasis and recurrence in various cancers [77]. In NSCLC, *CXCL5* plays a role in tumor growth, metastasis, and angiogenesis. An analysis showed that an upregulation of *CXCL5* expression was an independent predictor for poor prognosis and unfavorable response to ICIs. High expression of *CXCL5* was also associated with worse progression-free survival (PFS). This suggests that *CXCL5* may be a potential biomarker for prognosis and responsiveness to immunotherapy and could be a novel therapeutic target for NSCLC [78]. Targeting *CXCR2* has been shown to inhibit lung cancer progression and promote the therapeutic effect of cisplatin [79].

*CXCL8* (IL-8): Secreted primarily by macrophages, *CXCL8* has a proinflammatory function and binds to *CXCR1* and *CXCR2*. It stimulates cell proliferation in NSCLC through epidermal growth factor receptor transactivation. High levels of *CXCL8* mRNA in the lung TME correlate with increased angiogenesis and average survival in lung cancer. High serum levels of IL-8 in advanced NSCLC patients have been correlated with poor prognosis. *CXCL8* is considered a prospective biomarker to forecast tumor burden, treatment response, and survival. The ongoing clinical trial NCT04123379 is investigating Nivolumab with BMS-986253 (against *CXCL8*) to increase immune response against NSCLC [3,4,80].

#### 4.3.4. *CX3CR1/CX3CL1* (Fractalkine)

*CX3CR1*, the receptor for the chemokine *CX3CL1* (also known as fractalkine), has emerged as a potentially informative biomarker and functional player in antitumor immunity. The “*CX3CR1* score” has been explored as a dynamic, blood-based biomarker for predicting response to immune checkpoint inhibitors (ICIs). Recent findings in non-small cell lung cancer (NSCLC) patients demonstrated a higher proportion of *CX3CR1*^+^ CD8^+^ T cells in peripheral blood compared to tumor tissue. Although these *CX3CR1*^+^ CD8^+^ T cells may remain in circulation and not infiltrate the tumor microenvironment (TME) in the absence of *CX3CL1* production, their presence in peripheral blood may still serve as a valuable indicator of therapeutic response [81]. Additionally, studies in *CX3CR1*-deficient mice have revealed impaired antitumor responses, underscoring the importance of the *CX3CR1–CX3CL1* axis in mediating effective immune activity against tumors. [4]. Furthermore, bidirectional signaling between macrophages and cancer cells involving *CCR2* and *CX3CR1* has been identified as a key mechanism driving lung cancer progression, highlighting the broader immunological relevance of this axis beyond T cell activity [50].

#### 4.3.5. CCR9 and CCR10

Chemokine receptors such as *CCR9* and CCR10 have also been identified as potential biomarkers in non-small cell lung cancer (NSCLC). In particular, elevated pretreatment levels of circulating *CD4^+^CCR9*^+^ and *CD4^+^CCR10^+^* T cells have been associated with significantly poorer overall survival in patients with advanced lung cancer undergoing anti-PD-1 immune checkpoint inhibitor (ICI) therapy. Specifically, high levels of *CD4^+^CCR9^+^* T cells were linked to a median overall survival of 15.7 months compared to 35.9 months in patients with lower levels (hazard ratio [HR] 0.16, *p* = 0.003), while patients with elevated *CD4^+^CCR10^+^* T cells had a median survival of 22.0 months versus a non-reached median in those with low expression (HR 0.10, *p* = 0.003). Notably, this prognostic impact appeared to be specific to ICI-treated individuals [2,11].

From a biological standpoint, *CCR9* has been shown to facilitate the migration and invasion of lung adenocarcinoma cancer stem cells, while *CCR10* plays a crucial role in glioma progression by promoting proliferation, invasion, and affecting patient survival, thereby underscoring the broader functional significance of these chemokine receptors in cancer pathophysiology [4,11,40,44].

You can see a summary at Table 1.

## 5. Peripheral Blood Mononuclear Cells (PBMCs) as a Source of Biomarkers

The shift towards liquid biopsies and blood-based biomarkers is driven by the limitations of traditional tissue biopsies and the need for dynamic monitoring of treatment response [10].

Advantages of PBMCs:

Peripheral blood mononuclear cells (PBMCs) present several advantages that make them an invaluable tool in immunological research and clinical monitoring. They can be obtained through routine venipuncture, representing a minimally invasive, low-risk, and convenient method for repeated sampling. This enables longitudinal assessments, allowing for real-time monitoring of immune status and treatment response. Moreover, peripheral blood offers a comprehensive overview of the systemic immune landscape, which is a critical determinant of immunotherapy efficacy. PBMCs encompass a diverse array of circulating immune cell subsets—including T lymphocytes, B lymphocytes, natural killer cells, and monocytes—whose phenotypic and functional characteristics can provide meaningful insights into the patient’s immune response to cancer and therapeutic interventions [10,12,82,83,84,85].

Challenges and Future Directions for PBMC Biomarkers:

Despite their potential, the use of peripheral immune cells as predictive biomarkers faces several challenges that must be addressed to ensure clinical utility. One major limitation is the heterogeneity of findings across studies, which often stems from small sample sizes and methodological inconsistencies; for example, while some studies have associated higher baseline absolute lymphocyte counts with improved outcomes, others have reported conflicting results. Additionally, many of the candidate biomarkers identified thus far lack robust validation and require confirmation through prospectively designed studies involving larger and more diverse patient cohorts. To elucidate the mechanisms underlying *anti-PD-1*-mediated responses, functional assays will be essential to characterize specific PBMC subsets and their immunological activities. Furthermore, integrating PBMC immunophenotyping with established biomarkers—such as *PD-L1* expression or tumor mutational burden (TMB)—may offer a novel and potentially complementary strategy for selecting patients most likely to benefit from immunotherapy [2,10,11]. However, there are no studies or databases that offer information on the expression of chemokine receptors in peripheral immune system cells as biomarkers, beyond our data reported in the publications from 2022 and 2025 [2,11]. The most relevant data are found at the level of *CXCR4*, *CCR9* and *CCR10* expressed in PBMCs. These chemokines, studied by flow cytometry, have been shown to be expressed in cytotoxic (*CXCR4*) and helper (*CCR9* and *CCR10*) T cells and interfere with the efficacy of immunotherapy in NSCLC. In fact, it has been seen that patients with low *CXCR4* expression in cytotoxic T lymphocytes in peripheral blood have a better efficacy to anti-*PD-1* ICIs in NSCLC because probably they are receiving trafficking orders from a higher expression of *CXCL12* in tumor tissue [2,11].

## 6. Conclusions

NSCLC continues to be a major global health concern with high mortality rates, particularly in its advanced stages. ICIs have brought revolutionary advancements in treatment; the variability in patient response and the limitations of current tissue-based biomarkers underscore the critical need for more reliable and less invasive predictive tools.

This review highlights the burgeoning role of chemokine receptors expressed on PBMCs, especially in T cells, as promising prognostic and predictive biomarkers for immunotherapy efficacy in NSCLC (Table 1, Figure 1). The intricate interplay of chemokines and their receptors, which cancer cells often co-opt for their proliferation, survival, metastasis, and modulation of the tumor microenvironment, provides a rich source of potential targets for both diagnostics and therapeutics. Possibly, the integration in prospective studies of these biomarkers in peripheral blood with other already established biomarkers, such as TMB or PD-L1 expression in tumor tissue, will allow us to improve the selection of patients who will benefit most from ICIs in NSCLC and in other tumors.

In conclusion, we would like to rule out the following chemokine pathways as candidates for biomarkers or targets in lung cancer immunotherapy (Figure 1):(1)CXCR4 overexpression in tumor tissue is consistently associated with poorer outcomes in NSCLC, including increased metastasis to lymph nodes, brain, and pleural space, as well as enhanced tumor growth, invasion, and chemo-resistance. CXCR4 antagonists are actively being developed to counteract these pro-tumorigenic effects and to enhance the efficacy of immunotherapy by overcoming immune suppression. Crucially, studies on PBMCs suggest that high baseline levels of circulating *CD8+ CXCR4^+^* T cells are associated with poorer overall survival in immunotherapy-treated patients, indicating that the peripheral immune status regarding this receptor could be a significant predictive factor. Conversely, low circulating *CXCR4*-expressing cytotoxic T lymphocytes may correlate with increased tumor infiltration and better immunotherapy benefit [2,11].(2)CXCR6 plays a vital role in the migration and function of *CD8*^+^ resident memory T cells (TRM) in the lung mucosa, which are critical for effective antitumor immunity. Deficiency in *CXCR6* impairs the efficacy of cancer vaccines and responsiveness to anti-PD-1 treatment in preclinical models. The finding that *CXCR6* is significantly upregulated in *CD8*^+^ MAIT cells of immunotherapy responders further strengthens its potential as a predictive biomarker, with higher *CXCR6^+^ CD8^+^* MAIT cell ratios in peripheral blood correlating with better progression-free survival.(3)Other chemokine receptors, such as *CXCR3*, *CCR5*, *CXCR2*, *CCR9*, *CCR10*, and CX3CR1, also exhibit significant associations with NSCLC progression and response to immunotherapy. For instance, while intratumoral *CXCR3* activity is required for anti-PD-1 efficacy, high post-treatment *CXCL10* (*CXCR3* ligand) correlates with poor overall survival. High circulating levels of *CD4^+^ CCR9^+^* and *CD4^+^ CCR10^+^* T cells are linked to poorer outcomes in ICI-treated NSCLC patients. The *CXCL5/CXCR2* axis, characterized by neutrophil infiltration, is associated with unfavorable immunotherapy responses and poor prognosis. *CX3CR1* expression on circulating CD8^+^ T cells also holds promise as a dynamic blood-based biomarker.

The accessibility and non-invasive nature of PBMCs make them an ideal source for such biomarkers, allowing for real-time monitoring of treatment response and a more comprehensive understanding of the systemic immune status. However, the field is still evolving, and the heterogeneity of findings need further validation through large-scale, prospectively designed clinical studies. Moreover, detailed functional assays are crucial to fully elucidate the mechanisms by which these chemokine receptor-expressing PBMC populations influence immunotherapy outcomes.

## 7. Future Directions

In summary, understanding the expression patterns and functional roles of chemokine receptors in peripheral blood mononuclear cells offers a promising avenue for developing novel, non-invasive predictive biomarkers, in combination with PD-L1 or other biomarkers, such as TMB. This knowledge can significantly improve patient selection for ICI therapies, optimize treatment strategies, and ultimately improve the prognosis for patients with non-small cell lung cancer in the era of personalized medicine. Furthermore, since these biomarkers are easy to detect, they can help us understand the dynamics and effect of the ICIs in peripheral blood almost immediately.

## Figures and Tables

**Figure 1 curroncol-32-00583-f001:**
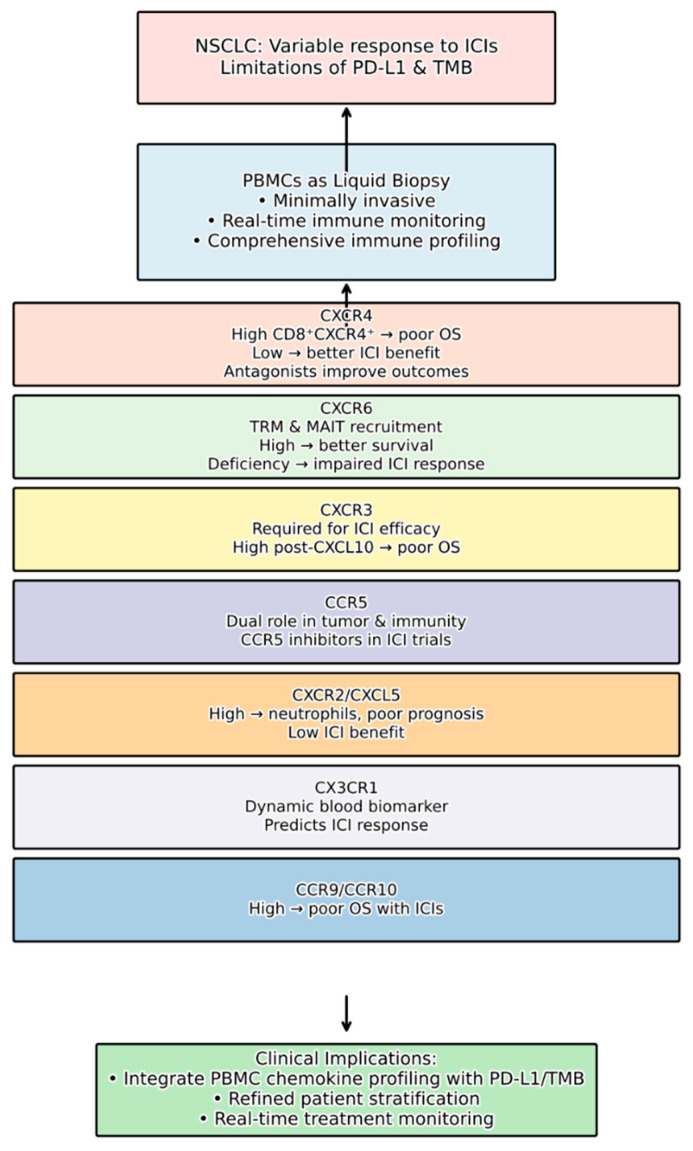
Peripheral Blood Chemokine Receptors as Predictive Biomarkers for Immunotherapy in NSCLC: application summary.

**Table 1 curroncol-32-00583-t001:** Peripheral Blood Chemokine Receptors as Predictive Biomarkers for Immunotherapy in NSCLC: application summary.

Chemokine Receptor	Ligand(s)	Role in NSCLC Immunobiology	Circulating PBMC Association	Prognostic/Predictive Impact	Therapeutic Targeting/Trials
**CXCR4**	CXCL12	Promotes tumor growth, metastasis, immune suppression	High CD8^+^CXCR4^+^ T cells: poor OS Low levels:better ICI benefit	Poor OS, especially in females; linked to advanced stage and adenocarcinoma	Antagonists: AMD3100, BKT140; synergistic with ICIs. No data in NSCLC
**CXCR6**	CXCL16	Guides CD8^+^ TRM and MAIT cell migration to lung; enhances local immunity	High CXCR6^+^ MAIT cells in responders	Better PFS and OS; deficiency impairs ICI efficacy	Potential biomarker for vaccine and ICI response. No trials in NSCLC
**CXCR3**	CXCL9, CXCL10, CXCL11	Effector T-cell recruitment to TME	Sustained high CXCL10 post-therapy poor outcome	Required for ICI efficacy intratumorally	None in routine use; experimental modulation
**CCR5**	CCL5	Tumor progression and immune regulation	Required in CD4^+^/CD8^+^ for maximal TME immune activity	Target in ongoing CCR2/CCR5 antagonist + ICI trials	None in routine use; experimental modulation
**CXCR2**	CXCL5, CXCL8	Neutrophil recruitment, angiogenesis, tumor progression	High CXCL5: poor prognosis, unfavorable ICI response	Independent poor prognostic factor	CXCR2 blockade synergizes with chemotherapy. No data in NSCLC
**CX3CR1**	CX3CL1	T cell–tumor interaction; potential “score” biomarker	Circulating CX3CR1^+^CD8^+^ predictive of ICI response	Emerging dynamic biomarker	None in standard care; under investigation
**CCR9/CCR10**	CCL25/CCL27–28	CSC migration (CCR9), glioma progression (CCR10)	High CD4^+^CCR9^+^/CCR10^+^ T cells poor OS with ICIs	Negative prognostic in ICI-treated patients	No targeted drugs yet

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
