# Peer review of "Chemokine Receptors in Peripheral Blood Mononuclear Cells as Predictive Biomarkers for Immunotherapy Efficacy in Non-Small Cell Lung Cancer"

_curroncol, 2025, doi:10.3390/curroncol32100583_

Round 1
Reviewer 1 Report
Comments and Suggestions for Authors
The manuscript “Chemokine Receptors in Peripheral Blood Mononuclear Cells as Predictive Biomarkers for Immunotherapy Efficacy in Non-Small Cell Lung Cancer” addresses a highly relevant and timely topic. The review is comprehensive, well-referenced, and clearly written. It consolidates current knowledge on chemokine receptors in PBMCs and their potential utility as predictive biomarkers in NSCLC immunotherapy. The subject is of clinical and translational importance, and the review has value for oncologists and immunologists. I have some minor suggestions: 1- The authors could strengthen the originality by highlighting underexplored aspects, such as the integration of chemokine receptor signatures with established biomarkers (PD-L1, TMB), or potential use in real-time monitoring of treatment response. 2- Maybe they could explore public databases of the NSCLC patients to address it or complement the manuscript.
Author Response
Comments and Suggestions for Authors
The manuscript “Chemokine Receptors in Peripheral Blood Mononuclear Cells as Predictive Biomarkers for Immunotherapy Efficacy in Non-Small Cell Lung Cancer” addresses a highly relevant and timely topic. The review is comprehensive, well-referenced, and clearly written. It consolidates current knowledge on chemokine receptors in PBMCs and their potential utility as predictive biomarkers in NSCLC immunotherapy. The subject is of clinical and translational importance, and the review has value for oncologists and immunologists. I have some minor suggestions:
Many thanks to reviewer 1 for the suggestions and contributions. We have made the suggested modifications to the manuscript. Below, we detail each suggestion.
- The authors could strengthen the originality by highlighting underexplored aspects, such as the integration of chemokine receptor signatures with established biomarkers (PD-L1, TMB), or potential use in real-time monitoring of treatment response.
We have made changes to the text to add more emphasis to this data.
- Maybe they could explore public databases of the NSCLC patients to address it or complement the manuscript.
We have introduced comments on this in the manuscripts. It should be noted that there are no published data from databases exploring the expresión of chemokine receptors in PBMCs in patients with lung cancer. Most studies talk about the expression of these molecules in tumor cells.
Reviewer 2 Report
Comments and Suggestions for Authors
The author focuses on recent advances in chemokine receptors in PBMCs and discusses the epidemiology of NSCLC, the current status of immunotherapy, and the functions of chemokine receptors. The logic of the discussion is clear. Overall, the review is well-structured but still requires further revision and refinement:
1. The section on the mechanisms by which chemokines can serve as biomarkers is currently superficial and should be expanded to explore these mechanisms in greater depth, rather than providing only a general overview.
2. The review summarizes the application of chemokine receptors as predictive biomarkers; however, the corresponding table should be supplemented to include the use of chemokines as prognostic biomarkers, as well as updates on relevant clinical trials.
3. The language in this review ought to be improved.
Author Response
Comments and Suggestions for Authors
The author focuses on recent advances in chemokine receptors in PBMCs and discusses the epidemiology of NSCLC, the current status of immunotherapy, and the functions of chemokine receptors. The logic of the discussion is clear. Overall, the review is well-structured but still requires further revision and refinement:
Many thanks to reviewer for the suggestions and contributions. We have made the suggested modifications to the manuscript. Below, we detail each suggestion.
- The section on the mechanisms by which chemokines can serve as biomarkers is currently superficial and should be expanded to explore these mechanisms in greater depth, rather than providing only a general overview.
Thank you very much for the suggestion. We've added more text to this section.
- The review summarizes the application of chemokine receptors as predictive biomarkers; however, the corresponding table should be supplemented to include the use of chemokines as prognostic biomarkers, as well as updates on relevant clinical trials.
There are no other studies, beyond those included in the table, that use chemokine expression in peripheral blood as biomarkers. There are no clinical trials that use them for this purpose. There are only clinical or preclinical studies that correlate them, and the two most recent and best-designed studies are referenced (refs. 2 and 10). However, we have made the table clearer for better understanding.
- The language in this review ought to be improved.
We have done a thorough review of the English and corrected all the errors. Thank you for the suggestion.
Reviewer 3 Report
Comments and Suggestions for Authors
Although the idea of reviewing our knowledge on chemokine receptors in PBMCs is interesting, the paper fails to provide a clear view for the readers.
Major suggestions:
- Non-Small Cell Lung Cancer: Epidemiology, Classification, and Current Treatment Approaches: This part of the review simplistically presents trivial knowledge. No need to include in the paper.
- Chemokines and Their Receptors in Cancer Progression and the Tumor Microenvironment: The groups of chemokines and receptors should be exhaustively presented in a table. The way these are briefly presented provides no clear view of the main subject of the review.
- Chemokine Receptors as Prognostic and Predictive Biomarkers in NSCLC: The Authors present some clinical-pathological data on cytokine receptor expression in NSCLC and their eventual role in metastasis and prognosis. The main question raised is where these receptors are expressed. On cancer cells, on TILs, TAMs,, fibroblasts ? The title of the paper states that the focus is PBMCs. Do the authors mean lymphocytes? What subpopulations of PBMCs express what cytokine receptors, and how do these affect clinical behavior? Authors should clearly present the data from published papers, provided these clearly state the immune cell type where the receptors were expressed and how they assessed their expression (immunhistochmistry? Flow cytometry?). If ELISA serum/plasma assays were applied, this should be mentioned.
- Overall, the paper should be shortened and focused on chemokine and chemokine receptor expression association with ICI efficacy, and provide in-depth data and a critical presentation of the included reference papers.
Author Response
Comments and Suggestions for Authors
Although the idea of reviewing our knowledge on chemokine receptors in PBMCs is interesting, the paper fails to provide a clear view for the readers.
Many thanks to reviewer for the suggestions and contributions. We have made the suggested modifications to the manuscript. Below, we detail each suggestion.
Major suggestions:
- Non-Small Cell Lung Cancer: Epidemiology, Classification, and Current Treatment Approaches: This part of the review simplistically presents trivial knowledge. No need to include in the paper.
We have simplified this part. Thanks so much for the suggestion.
- Chemokines and Their Receptors in Cancer Progression and the Tumor Microenvironment: The groups of chemokines and receptors should be exhaustively presented in a table. The way these are briefly presented provides no clear view of the main subject of the review.
In the table we include the main chemokine target groups. This table has been clarified for easier understanding.
- Chemokine Receptors as Prognostic and Predictive Biomarkers in NSCLC: The Authors present some clinical-pathological data on cytokine receptor expression in NSCLC and their eventual role in metastasis and prognosis. The main question raised is where these receptors are expressed. On cancer cells, on TILs, TAMs,, fibroblasts ? The title of the paper states that the focus is PBMCs. Do the authors mean lymphocytes? What subpopulations of PBMCs express what cytokine receptors, and how do these affect clinical behavior? Authors should clearly present the data from published papers, provided these clearly state the immune cell type where the receptors were expressed and how they assessed their expression (immunhistochmistry? Flow cytometry?). If ELISA serum/plasma assays were applied, this should be mentioned.
We have clarified in the text that chemokine expression in PBMCs is observed in lymphocytes (in the mosto of the studies revised) and is detected by flow cytometry. We have expanded the publications, although there is little evidence, and have referenced and commented on some relevant articles from our group, among others.
Thank you very much for the suggestions; we believe these changes will enhance the manuscript's value.
- Overall, the paper should be shortened and focused on chemokine and chemokine receptor expression association with ICI efficacy, and provide in-depth data and a critical presentation of the included reference papers.
Thanks for the suggestion. We have shortened the manuscript and added more detail on what was mentioned. Thank you very much for the suggestion.
Round 2
Reviewer 1 Report
Comments and Suggestions for Authors
The authors have appropriately addressed previous comments and suggestions. Now, the manuscript could be published. Congrats
Reviewer 2 Report
Comments and Suggestions for Authors
- This review has refined the use of chemokine receptors as predictive and prognostic biomarkers.
- The mechanisms underlying the use of chemokine receptors as biomarkers have been thoroughly investigated.
Reviewer 3 Report
Comments and Suggestions for Authors
I have no further suggestions